# Comparison of Ultrafast Dynamic Contrast-Enhanced (DCE) MRI with Conventional DCE MRI in the Morphological Assessment of Malignant Breast Lesions

**DOI:** 10.3390/diagnostics13061105

**Published:** 2023-03-15

**Authors:** Akane Ohashi, Masako Kataoka, Mami Iima, Maya Honda, Rie Ota, Yuta Urushibata, Marcel Dominik Nickel, Masakazu Toi, Sophia Zackrisson, Yuji Nakamoto

**Affiliations:** 1Department of Translational Medicine, Diagnostic Radiology, Lund University, 225 02 Malmö, Sweden; 2Department of Imaging and Functional Medicine, Skåne University Hospital, 225 02 Malmö, Sweden; 3Department of Diagnostic Imaging and Nuclear Medicine, Kyoto University Graduate School of Medicine, Kyoto 606-8507, Japan; 4Institute for Advancement of Clinical and Translational Science (iACT), Kyoto University Hospital, Kyoto 606-8507, Japan; 5Department of Diagnostic Radiology, Kansai Electric Power Hospital, Osaka 553-0003, Japan; 6Department of Radiology, Tenri Hospital, Nara 632-8552, Japan; 7Siemens Healthcare K.K., Shinagawa, Tokyo 141-8644, Japan; 8MR Application Predevelopment, Siemens Healthcare GmbH, 91050 Erlangen, Germany; 9Department of Breast Surgery, Kyoto University Graduate School of Medicine, Kyoto 606-8507, Japan

**Keywords:** ultrafast DCE-MRI, breast MRI, morphology

## Abstract

Ultrafast (UF) dynamic contrast-enhanced (DCE)-MRI offers the potential for a faster and, therefore, less expensive examination of breast lesions; however, there are no reports that have evaluated whether UF DCE-MRI can be used the same as conventional DCE-MRI in the reading of morphological information. This study evaluated the agreement in morphological information obtained from malignant breast mass lesions between UF DCE-MRI and conventional DCE-MRI. UF DCE-MRI data were obtained over the first 60 s post-contrast injection, followed by the conventional DCE images. Two readers evaluated the size and morphology of the lesions in the final phase of the UF DCE-MRI and the early phase of the conventional DCE-MRI. Inter-method agreement in morphological information was evaluated for the two readers using the intraclass correlation coefficient for size, and the kappa statistics for the morphological descriptors. Differences in the proportion of each descriptor were examined using Fisher’s test of independence. Most inter-method agreements were higher than substantial. UF DCE-MRI showed a circumscribed margin and homogeneous enhancement more often than conventional imaging. However, the percentages of readings showing the same morphology assessment between the UF DCE-MRI and conventional DCE-MRI were 71.2% (136/191) for Reader 1 and 69.1% (132/191) for Reader 2. We conclude that UF DCE-MRI may replace conventional DCE-MRI to evaluate the morphological information of malignant breast mass lesions.

## 1. Introduction

Ultrafast dynamic contrast-enhanced (UF DCE) MRI is a novel acquisition approach that can reveal kinetic information from the very early phase of DCE-MRI, doing so with high spatial and temporal resolution. The usual temporal resolution of UF DCE-MRI is less than 10 s per frame, which is faster than the 60 s per frame of conventional DCE MRI [1]. Several accelerated imaging acquisition techniques are available for obtaining high spatial and temporal resolution including parallel imaging [2,3], view sharing [4,5], and compressed sensing (CS) [6,7]. The CS technique allows high spatial resolution by reconstructing images from randomly and highly undersampled k-space data acquired with a high temporal resolution [8]. With the development of the CS technique, high temporal resolution in the order of several seconds per frame, combined with a high spatial resolution of one-millimeter in-plane, can be achieved simultaneously [9]. The high temporal resolution allows us to obtain detailed information on the upslope portion of the kinetic curve. Several important kinetic parameters can be acquired from UF DCE-MRI including maximum slope (MS), defined as the upslope of the time–intensity curve; time to enhance (TTE), defined as the time when a breast lesion starts to enhance versus the time when the aorta starts to enhance; and bolus arrival time (BAT), defined as the time from the start of contrast injection to the arrival of the tracer bolus. These new parameters obtainable from UF DCE-MRI represent the uptake behavior of the contrast agent demonstrated by the time–intensity curve [9]. High MS, short TTE, and short BAT tend to indicate malignant lesions. Kinetic parameters extracted from UF DCE-MRI show equivalent or higher diagnostic performance to those from conventional DCE-MRI when used to differentiate benign from malignant breast lesions [10,11,12]. These results indicate that UF DCE-MRI has the potential to replace, or at least complement, conventional DCE-MRI in the diagnosis of breast lesions. There are mainly two advantages to introducing UF DCE-MRI. The first is the avoidance of breast parenchymal enhancement (BPE). UF DCE-MRI includes a very early dynamic phase acquired before BPE appears, which may help to detect cancer and mask other lesions that enhance conventional DCE-MRI. A large multicenter prospective study showed that MRI with moderate or significant BPE was associated with higher abnormal interpretation and biopsy rates and lower specificity than MRI, with minimal or mild BPE [13]. Considering this undesirable impact of BPE, reducing its effect by using UF DCE-MRI could be especially advantageous for lesion detection when evaluating breast lesions in premenopausal women or women with high BPE [14]. Second, UF DCE-MRI allows for much shorter scanning times and can be considered for use as part of an abbreviated MRI protocol for breast MRI screening [15,16]. Therefore, it is important to verify the diagnostic capabilities of UF DCE-MRI compared to the conventional DCE-MRI so that it can be widely used in future clinical practice and screening.

The Breast Imaging Reporting and Data System (BI-RADS) descriptors are widely used diagnostic descriptors of breast lesions [17]. BI-RADS recommends that lesions are evaluated using both kinetic and morphological information. Several reports have evaluated the diagnostic ability of UF DCE-MRI with a focus on kinematic information. Still, fewer studies have included morphological information in the evaluation of the diagnostic ability of UF DCE-MRI [18,19]. If the morphological assessment on UF DCE-MRI shows a high agreement rate with that on conventional DCE-MRI, it would be a significant advantage in facilitating the replacement of conventional DCE-MRI with UF sequences. Therefore, we aimed to evaluate the agreement in the morphological descriptors between UF DCE-MRI and the early phase of DCE-MRI. In this analysis, we focused on the evaluation of mass lesions, which represent the majority of target lesions in clinical practice.

## 2. Materials and Methods

### 2.1. Study Population

Our institutional review board approved this study protocol and waived the requirement for informed consent because of the retrospective study design. The study population consisted of female patients who underwent breast MRI including UF DCE-MRI and conventional DCE-MRI protocols, in our hospital between December 2015 and May 2020. UF DCE-MRI was performed on patients with known lesions detected on other imaging modalities (mammography and US), patients who showed marked BPE on previous MRI, and patients aged ≤50 years. Mass lesions that were histopathologically diagnosed as invasive breast cancers were included. The study was performed on a per-breast basis. If a single patient had multiple lesions on one side of the breast, the lesion that appeared most malignant or the largest lesion was selected as the index lesion for that breast. Exclusion criteria included the receipt of neoadjuvant chemotherapy or neoadjuvant endocrine therapy before breast MRI and failed or incomplete UF DCE acquisition due to technical problems or the patient’s condition.

According to the criteria above-mentioned, a total of 191 lesions in 187 patients (age range 31–87 years, mean age of 58 years) were included in the analysis. Among them, four patients had bilateral breast lesions, and these lesions were analyzed separately. A flowchart of the patient selection process is shown in Figure 1. The patient population used in this study was analyzed and previously reported [20]. This prior study aimed to explore the diagnostic potential of multiparametric MRI to predict malignant and benign breast lesions using parameters from UF DCE-MRI. In contrast, the current study compared the morphological descriptors from UF DCE-MRI with those from conventional DCE-MRI.

### 2.2. MRI Protocols 

All patients underwent breast MRI examinations in the prone position with a 3-T scanner (MAGNETOM Skyra or Prisma; Siemens Healthcare, Erlangen, Germany) using dedicated 16- or 18-channel bilateral breast coils. The standard protocols at our institution include T2-weighted images (axial orientation; 2D-turbo spin echo with fat suppression; repetition time/echo time [TR/TE], 5660/79 ms; flip angle [FA] 140°; field of view [FOV], 330 × 330 mm; matrix, 448 × 336; thickness, 3.0 mm), non-contrast T1-weighted images (axial orientation; volumetric interpolated breath-hold examination [VIBE]; TR/TE, 5.14/2.46 ms; FA 15°; FOV, 330 × 330 mm; matrix, 384 × 319; thickness, 2.5 mm), and diffusion-weighted MRI (axial orientation; single-shot echo planar imaging [EPI]). 

A fat-suppressed T1-weighted sequence (TR/TE, 3.8/1.4 ms; FA 15°; FOV, 330 × 330 mm; voxel resolution, 0.86 × 0.86 mm; matrix, 384 × 384; thickness, 1.0 mm) was used for three-time frames of the conventional DCE imaging. One pre-contrast and two post-contrast (initial phase: 1–2 min after the contrast injection; and delayed phase: 5–6 min after the contrast injection) conventional DCE images were acquired at a rate of 60 s/frame immediately after the UF DCE acquisition (there was no overlap between UF DCE-MRI and the initial phase of the conventional DCE MRI). In addition, high-resolution contrast-enhanced images were acquired between the initial and delayed phases of the DCE images, at 2–5 min after the contrast injection (both breasts; coronal orientation; three-dimensional volumetric interpolated breath-hold examination [3D-VIBE] with fat suppression; repetition TR/TE = 4.61/1.80 ms; FA, 15°; FOV, 330 × 330 mm; matrix, 512 × 461; thickness, 0.8 mm; total acquisition time, 2 min 26 s). The total scanning time, including the UF DCE-MRI, was similar to our standard protocol for conventional DCE-MRI.

UF DCE imaging was obtained using a non-standard research-focused compressed sensing 3D-gradient-echo volumetric interpolated breath-hold examination (CS-VIBE) sequence acquired from 15 s before the contrast injection to 60 s after contrast injection, with 2 s of preparation time followed by 20 frames acquired at a rate of 3.7 s/frame immediately after gadolinium injection (TR/TE 5.0–4.8/2.5 ms; FA 15°; 0.94 × 0.94 mm, matrix size 384 × 384, thickness 2.5 mm). First, a dose of 0.1 mL/kg gadobutrol contrast agent (Gadovist; Bayer AG, Leverkusen, Germany) was injected at a speed of 2.0 mL/s into the antecubital vein with a 22G intravenous catheter, then a 20-mL saline bolus was administered at the same rate. A power injector was used (Sonic Shot^®^; Nemoto Kyorindo, Tokyo, Japan) for the injection of the contrast agent and saline. CS reconstruction was performed in 30 iterations, with convergence speeds evaluated separately and retrospectively [21]. Detailed scan protocols for the UF DCE-MRI and conventional DCE-MRI are provided in Table 1. A schematic drawing of the order of our dynamic protocols including UF DCE-MRI is shown in Figure 2.

### 2.3. Image Analysis

Subtraction images were computed from the 20 frames of the original UF DCE-MR images by subtracting the first phase from each of the 2nd–20th phases. Then, from these subtraction images, 19 maximum intensity projection (MIP) images for each breast were reconstructed using a workstation (Aquarius NET Viewer; TeraRecon, Foster City, CA, USA). To evaluate the consistency of the visual assessments in terms of morphology, two radiologists (Readers 1 and 2, with 8 and 23 years of experience in diagnosing breast MRI, respectively [A.O. and M.K.]) were involved as readers to evaluate the size and morphological descriptors of the lesions (shape: round, oval, or irregular; margin: circumscribed, irregular, or spiculated; and internal enhancement: homogeneous, heterogeneous, or rim enhancement) on the final phase (20th phase) of the UF DCE-MRI and the early phase of DCE-MRI. The morphological descriptors were based on MRI BI-RADS (Breast Imaging Reporting and Data System) by ACR (American College of Radiology), 5th edition, 2013 [17].

### 2.4. Histopathological Evaluation

All lesions were histopathologically confirmed by biopsy before surgical resection. One dedicated pathologist (over 20 years of experience) retrospectively reviewed the histopathology results using the World Health Organization classification of breast tumors [22]. The variables extracted from the histological information were tumor histology, histopathological grade, Ki-67, and immunohistochemistry status, including estrogen receptor (ER), progesterone receptor (PR), and human epidermal growth factor receptor type 2-enrichment (HER-2). Invasive carcinomas were categorized as luminal A-like (ER+ and/or PR+, HER2-, Ki-67 < 14%), luminal B-like (ER+ and/or PR+, HER2+ or −, Ki-67 > 14%), HER-2 (ER/PR−, HER2+), or triple-negative. The detailed pathological diagnoses and subtypes of the lesions included in this study are shown in Table 2.

### 2.5. Statistical Analysis

The inter-method agreements in the morphological information between UF DCE-MRI and conventional DCE-MRI were evaluated for Readers 1 and 2 using the intraclass correlation coefficient for size and the Cohen’s kappa statistics for the morphological descriptors. Kappa statistics were calculated using the scale reported by Viera and Garre [23].

The inter-reader agreements between Readers 1 and 2 were also evaluated to ensure reasonable reproducibility on UF DCE-MRI, with this analysis using the intraclass correlation coefficient for size and the Cohen’s kappa statistics for the morphological descriptors. The inter-reader agreements on the conventional DCE-MRI were calculated for the purpose of comparison.

The intraclass correlation coefficients were interpreted according to the following criteria: ≤0.40 indicated poor agreement, 0.40 to 0.59 indicated fair agreement, 0.60 to 0.74 indicated good agreement, and 0.75 to 1.00 indicated excellent agreement. Kappa values were interpreted as: 0.01 to 0.20 indicated slight agreement, 0.21 to 0.40 indicated fair agreement, 0.41 to 0.60 indicated moderate agreement, 0.61 to 0.80 indicated substantial agreement, and 0.81 to 1.00 indicated almost perfect agreement.

Differences in the proportion of each descriptor between UF DCE-MRI and conventional DCE-MRI were examined using Fisher’s test of independence. Statistical analyses were performed with EZR (Saitama Medical Center, Jichi Medical University) [24].

## 3. Results

### 3.1. Inter-Methods Agreement in Morphological Information between UF DCE-MRI and the Early Phase of DCE-MRI

The inter-method agreements in the morphology and size between ultrafast DCE-MRI and conventional DCE-MRI are shown for the two readers in Table 3. Most inter-method agreements between UF DCE-MRI and the early phase of DCE-MRI were higher than substantial for both readers. The agreements for internal enhancement were substantial or moderate, indicating that UF DCE-MRI images are not sufficient for evaluating it, although the size, shape, and margins may be judged similarly to conventional DCE-MRI.

### 3.2. Inter-Reader Agreements for Morphological Information

The inter-reader agreements between Readers 1 and 2 for size and morphology on UF DCE-MRI were: intraclass correlation coefficient of 0.998 (95% CI 0.997–0.998) for lesion size, representing almost perfect agreement, and kappa values of 0.821 (95% CI 0.792–0.841) for shape, 0.942 (95% CI 0.934–0.954) for margin, and 0.846 (95% CI 0.778–0.885) for internal enhancement, each representing almost perfect agreement. The agreements on conventional DCE-MRI between the readers were: intraclass correlation coefficient of 0.997 (95% CI 0.996–0.998) for lesion size, representing almost perfect agreement, and kappa values of 0.782 (95% CI 0.737–0.804) for shape, 0.691 (95% CI 0.644–0.753) for margin, and 0.834 (95% CI 0.809–0.853) for internal enhancement, with the first two values representing substantial agreement and the latter almost perfect agreement. A high concordance rate was obtained, even though the radiologists had different experience levels. The inter-reader agreements for UF DCE-MRI were higher than those of conventional DCE-MRI.

### 3.3. Details of the Morphological Diagnosis

Detailed results of the morphological assessments on UF DCE-MRI and conventional DCE-MRI are shown in Table 4. UF DCE-MRI showed a more circumscribed margin, more homogeneous enhancement, and less rim enhancement (morphology assessments were more likely to be benign) than the conventional DCE-MRI. In addition, the frequency of internal enhancement was significantly different between the two acquisitions (*p* < 0.001).

The percentages of readings showing the same morphology assessment between the two methods were 71.2% (136/191) for Reader 1 and 69.1% (132/191) for Reader 2. Table 5 shows a breakdown of those cases in which the morphological diagnosis was discrepant between the two acquisitions. Some of the results on conventional DCE-MRI tended to show more findings of “suspicious for malignancy” as the correct diagnosis compared with UF DCE-MRI. For example, an irregular margin on conventional DCE MRI tended to show as a circumscribed margin on UF DCE-MRI, and heterogeneous or rim enhancement on conventional DCE-MRI tended to show as a homogeneous enhancement on UF DCE-MRI. Representative images showing the same morphology and different morphology between UF DCE-MRI and conventional DCE-MRI are shown in Figure 3. In the first case, the morphology findings were an irregular margin and homogeneous enhancement on both scans. The second case was judged as benign on the UF DCE-MRI but as malignant with an irregular margin and heterogeneous enhancement on the conventional DCE-MRI.

## 4. Discussion

This study demonstrated that inter-method agreements for morphological assessments between UF DCE-MRI and the early phase of DCE-MRI were substantial, almost perfect, or excellent, for both Readers 1 and 2. The results indicate that UF DCE-MRI can potentially be used to evaluate the morphological features of breast lesions using the same criteria as conventional DCE-MRI. In particular, lesion size and shape demonstrated excellent or almost perfect agreement between UF DCE-MRI and the early phase of DCE-MRI. When we investigated the details of the lesions with discrepant evaluations, mass lesions on the UF DCE-MRI tended to be described as having a circumscribed margin or homogeneous internal enhancement patterns. Our results showed sufficient agreement in the morphological descriptors between the UF DCE-MRI and conventional DCE-MRI, with 71.2% (136/191) of the morphological diagnoses being the same for both acquisitions.

In addition, inter-reader agreement for the shape, margin, and internal enhancement was higher for UF DCE-MRI (being almost perfect, with kappa values of 0.821, 0.942, and 0.846, respectively) than for conventional DCE-MRI (kappa values: 0.782, 0.691, 0.834, respectively).

DCE-MRI generally obtains the highest sensitivity of all breast imaging modalities [25]. It is frequently used for breast cancer screening in high-risk women, for staging, and for evaluation after chemotherapy. For women with a high risk of developing breast cancer, supplemental annual MRI screening is recommended [26]. This is because they tend to develop breast cancer at a younger age than women with an average risk [27], and their breast cancers generally have a higher Nottingham histological grade and show rapid progression [28]. Recent studies have shown that in high-risk populations, especially women with extremely dense breasts, additional MRI screening can significantly reduce the interval cancer rates compared with mammography alone [29]. However, the high costs and limited availability of breast MRI are important issues. To make breast MRI more widely available, its cost-effectiveness must be improved, which involves shortening the protocols. After Kuhl et al. introduced an abbreviated protocol to shorten the scanning time [15], several studies reported different abbreviated protocols focusing on the detection of breast lesions [16]. UF DCE-MRI is a high-speed imaging technique that allows for the early visualization of the contrast inflow into the lesion. UF DCE-MRI can be added to abbreviated MRI protocols. Because of its shorter scanning time, UF DCE-MRI can be acquired with lower financial costs and time requirements than conventional DCE-MRI, and it could therefore be an important technique for MRI-based screening. For UF DCE-MRI to be realistically used in clinical practice, it is necessary to present the possibility that it can be used for the same assessments as conventional DCE-MRI.

BI-RADS is a commonly used image interpretation method because of the effectiveness of the diagnostic criteria combining kinetic and morphological information [17]. Kinkeil et al. demonstrated that margin morphology (*p* = 0.001) and enhancement pattern (*p* = 0.001) were the most significant parameters for lesion characterization [30]. Tozaki et al. evaluated an interpretation model using BI-RADS 4th edition descriptor based on the morphologic features of focal masses [31]. The lesion shapes/margins of malignant lesions were often irregular (47%) and spiculated (43%), whereas benign lesions more frequently showed smooth and lobular margins (93%). In addition, the most frequent feature of lobular-shaped malignant lesions was a washout kinetic pattern. The presence of rim enhancement has been associated with malignancy. Kuwada et al. reported that rim enhancement in the delayed phase, followed by central washout, was a feature of malignancy [32]. In our study, some lesions did not show visible rim enhancement on UF DCE-MRI, with it then becoming visible only in the early phase of conventional DCE-MRI. This might be related to a phenomenon similar to that in the report of Kuwada et al., which would be a limitation of UF DCE-MRI. Our previous study compared a conventional kinetic pattern of washout with that on UF DCE-MRI (MS was a kinetic parameter), and MS provided a similar diagnostic performance to a conventional washout-kinetic pattern [11]. If the morphology and kinetic patterns extracted from UF DCE-MRI have high diagnostic performance, they will be valuable for diagnostic use.

Several studies have reported that in comparison with conventional DCE-MRI, UF DCE-MRI-derived kinetic parameters had equal or better diagnostic ability for differentiating malignancy [10,11,12]. Zelst et al. evaluated the diagnostic performance of UF DCE-MRI breast screening using BI-RADS categories in a multileader study [33], and Dalmis et al. evaluated the diagnostic performance of a multiparametric UF DCE-MRI protocol using artificial intelligence techniques [34]. Although they found that the classification of benign and malignant lesions using UF DCE-MRI was non-inferior to conventional MRI, they did not specifically evaluate the differences in the accuracy or specific morphologic features. A morphological evaluation is needed when UF DCE-MRI is used for further classification (e.g., between malignant diseases). However, there are no reports that have evaluated whether UF DCE-MRI can be used in the same way as conventional DCE-MRI in the reading of morphological information. Therefore, our report is the first study to assess the morphological descriptors of breast lesions on UF DCE-MRI.

A previous study compared the diagnostic performance of a kinetic parameter derived from UF DCE-MRI using K-spatial-weighted image contrast (KWIC) with the BI-RADS category of conventional DCE-MRI. It was found that the BI-RADS category achieved higher sensitivity than the kinetic information from UF DCE-MRI [11]. This result may be because the BI-RADS category includes morphological information as well as kinetic information, whereas the KWIC acquisition did not have sufficient spatial resolution to evaluate the morphology. UF DCE-MRI using CS-VIBE can produce images with a higher spatial resolution than KWIC [7], and therefore, UF DCE-MRI should allow for a more accurate assessment of the morphological information, which led us to perform this study. The present results suggest that UF DCE-MRI using the CS technique may be used as an alternative to conventional DCE-MRI in the morphological evaluation of malignant mass lesions. Although many studies on UF DCE-MRI have focused on kinetic parameters and their use in diagnosis and prognostication [35,36], morphological information can be added to further improve the performance of UF DCE-MRI. For example, intra-ductal papilloma and fibroadenoma lesions with high blood flow are more likely to show fast and early inflow in UF DCE-MRI and be wrongly diagnosed as malignant (falsely positive), but if a morphological diagnosis can also be made using UF DCE-MRI, it may be possible to determine benignity and malignancy correctly.

During the past decade, morphological information has been used not only for predicting malignancy but also to evaluate the molecular subtypes of breast cancer. The morphology descriptor of rim enhancement is known to be highly associated with triple-negative breast cancer (TNBC) [37,38]. Uematsu et al. reported that rim enhancement was present in 80% of TNBCs and that rim enhancement was associated with smooth margins [37]. Alexander et al. found that rim enhancement on DCE-MRI was associated with long-term outcomes of patients with TNBC [38]. Seyfettin et al. evaluated the MRI features of molecular subtypes and found that luminal subtypes showing as irregular in shape and with an irregular/spiculated margin had a better prognosis. This was attributed to a desmoplastic response, in which less-aggressive slower-growing lower-stage tumors in the surrounding tissues suppressed tumor growth [39]. In comparison, the TNBC and HER2 subtypes tend to show mostly oval/round shapes and smooth margins [40]. Because of their rapid growth pattern, these tumors do not have the time to infiltrate the surrounding tissue, and the margin expands by pushing the surrounding tissue [41]. The differentiation of subtypes according to morphological diagnosis presents new possibilities for imaging biomarkers and prognostication.

We also identified limitations in the morphological information obtained from UF DCE-MRI. In some cases, the morphology on UF DCE-MRI tended to be more circumscribed, less spiculated, and more homogeneous with less rim enhancement, all of which represent benign characteristics. Two reasons can be considered for these differences. First, it may be that the timing of the UF DCE-MRI acquisition was too early to delineate details of the internal enhancement of the lesion because heterogeneity within the lesion may not become apparent until the contrast agent has been partially washed out from the lesion; it is challenging to detect heterogeneity within a lesion when it is filled with a contrast agent in the very early phase of contrast enhancement. The other reason is that the spatial resolution of UF DCE-MRI may still occasionally be insufficient to assess certain morphological characteristics. This is evidenced in some cases where homogeneous internal enhancement was changed to heterogeneous enhancement or rim enhancement on conventional DCE-MRI, as shown in Table 5. In contrast, the fact that mass lesions tend to be more likely to present with a circumscribed margin and homogeneous enhancement on UF DCE-MRI may result in better consistency between readers.

There were several limitations to our study. First, this study was of a retrospective design and used a relatively small dataset from a single medical center. Second, the analysis was inadequate for a clinical situation because we focused only on mass lesions, and non-mass enhancement was excluded, as mentioned in the introduction. A recent study evaluated the size and morphology of ductal carcinoma in situ (DCIS) using UF DCE-MRI with the CS technique. DCIS mostly takes the form of a non-mass lesion, and the study found that the lesion size tended to be smaller and that a clustered ring was not frequently observed on UF DCE-MRI compared to conventional DCE-MRI [42]. This result implies that caution is needed when evaluating DCIS on UF DCE-MRI, that the evidence from the morphological evaluation of mass and non-mass lesions using UF DCE-MRI is insufficient, and that further evaluations are needed. Third, the study population excluded patients with non-invasive carcinoma and benign lesions, and the lesion distribution therefore differs from that in normal clinical settings. Further studies including all variables are needed. Fourth, the kinetic behavior of gadobutrol may be slightly different to that of other contrast agents [43,44]. However, we were not able to examine the differences in UF DCE-MRI according to different contrast media or devices. The universal use of this acquisition is a future consideration. Fifth, because some malignant lesions may appear benign when diagnosed according to morphology alone, future studies should consider the diagnostic evaluation of malignant and benign lesions in combination with the kinetic information from CS-VIBE. Finally, our study used visual independent observer ratings, and since automatic texture determination was not performed, the results may be dependent on the experience and ability of the observer.

## 5. Conclusions

We have shown that UF DCE-MRI has the potential to be used for the evaluation of the morphological information of malignant mass lesions, performing as well as conventional DCE-MRI. Our results demonstrate the usefulness of UF DCE-MRI and should help ensure the accuracy of future screening protocols using only UF DCE-MRI as contrast imaging for breast MRI.

## Figures and Tables

**Figure 1 diagnostics-13-01105-f001:**
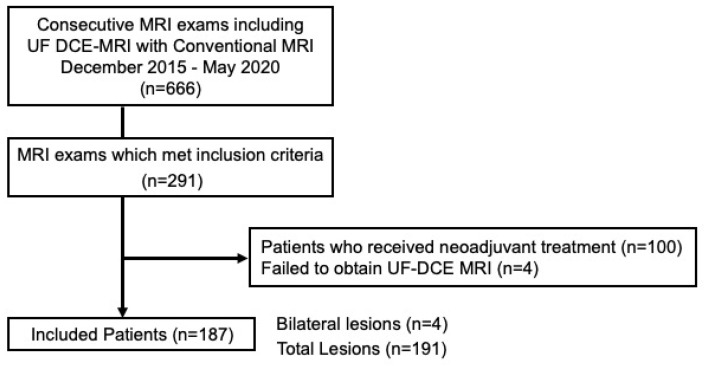
Flow diagram of the study population. UF DCE Ultrafast dynamic contrast-enhanced.

**Figure 2 diagnostics-13-01105-f002:**
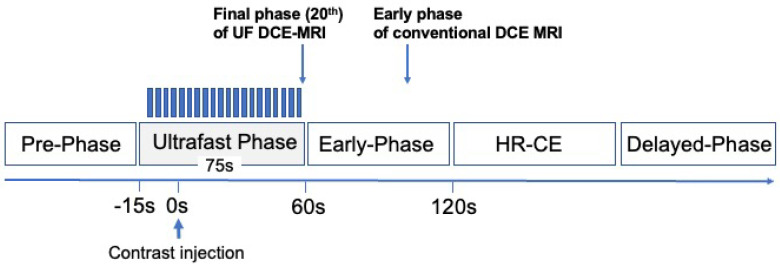
Schematic explanation of our dynamic protocols, including the UF DCE−MRI. UF DCE−MRI using CS−VIBE: 20 frames (pre + 19 frames), 3.7 s/frame. UF, ultrafast; VIBE, volumetric interpolated breath−hold examination; HR, high−resolution.

**Figure 3 diagnostics-13-01105-f003:**
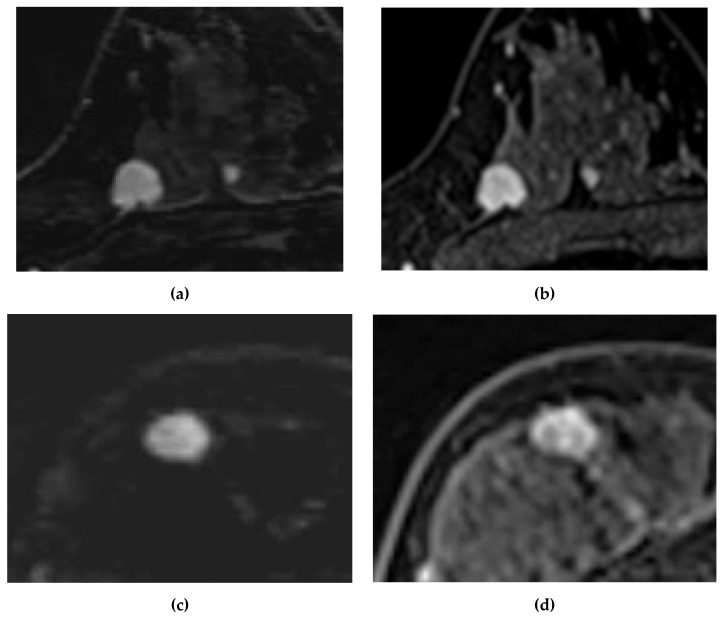
Cases showing agreements and discrepancies in morphology between UF DCE-MRI and conventional DCE-MRI. Case 1: A 49-year-old woman with invasive ductal carcinoma. Case 2: A 43-year-old woman with invasive ductal carcinoma. (**a**) UF DCE-MRI of Case 1. (**b**) Conventional DCE-MRI of Case 1. (**c**) UF DCE-MRI of Case 2. (**d**) Conventional DCE-MRI of Case 2. Both images of Case 1 show the same morphology with a circumscribed margin and homogeneous enhancement. However, (**c**) shows a circumscribed margin and homogeneous enhancement, whereas (**d**) shows an irregular margin and rim enhancement; (**c**) would likely result in a diagnosis of a benign lesion.

**Table 1 diagnostics-13-01105-t001:** Scan protocols of the conventional DCE-MRI and UF DCE-MRI.

	Conventional DCE-MRI	UF DCE-MRI
Sequence	VIBE with FS	VIBE without FS
TR/TE (ms)	3.8/1.4	5.0/2.5 *^1^, 4.8/2.5 *^2^
Flip Angle (degree)	15	15
Slice thickness (mm)	1.0	2.5
FOV (mm^2^)	330 × 330	360 × 360
Resolution (mm^2^)	0.86 × 0.86 (144 slices)	0.94 × 0.94 (60 slices)
Matrix	384 × 384 *^3^	384 × 384 *^4^
Fat-suppression	SPAIR	-
Temporal resolution (s/frames)	60 [3 frames: pre, early, delay]	3.7 [20 frames]
Total acquisition time (s)	180	75

VIBE, volumetric interpolated breath-hold examination; FS, fat suppression; TR/TE, repetition time/echo time; FOV, field of view; SPAIR, spectral attenuated inversion recovery. *^1^ MAGNETOM Skyra, *^2^ MAGNETOM Prisma, *^3^ The phase direction data were obtained with a matrix of 384 × 302, with partial Fourier acquisition of 7/8 and 90% phase resolution, then reconstruction into a matrix of 384 × 384, *^4^ The phase direction data were obtained with a matrix of 384 × 269 with 70% phase resolution and then reconstructed into a matrix of 384 × 384.

**Table 2 diagnostics-13-01105-t002:** Pathological diagnoses of breast cancers (n = 191).

Pathological Diagnosis	Case Number	LA	LB	HER-2	TNBC
NST	162	58	72	9	23
ILC	8	4	4	0	0
Mucinous carcinoma	7	4	2	1	0
Invasive micropapillary carcinoma	3	2	0	0	1
Invasive carcinoma with apocrine differentiation *	6	0	0	0	6
Histiocytoid carcinoma	2	0	0	0	2
Other	3	2	0	0	1
Total	191	70	78	10	33

NST, invasive ductal carcinoma of no special type; ILC, invasive lobular carcinoma; LA, luminal A-like; LB, luminal B-like; HER-2, epidermal growth factor receptor type 2; TNBC, triple-negative breast cancer. * Including one lesion with a metaplastic component.

**Table 3 diagnostics-13-01105-t003:** Agreements in the size and morphology between ultrafast DCE-MRI and conventional DCE-MRI for Readers 1 and 2.

Size and Morphology	ICC/Kappa	95% CI	Agreement
Reader 1			
Size	1.00	0.995–0.997	Excellent
Shape	0.95	0.913–0.994	Almost perfect
Margin	0.71	0.590–0.828	Substantial
Internal enhancement	0.61	0.523–0.704	Substantial
Reader 2			
Size	1.00	0.996–0.998	Excellent
Shape	0.95	0.905–0.994	Almost perfect
Margin	0.62	0.476–0.771	Substantial
Internal enhancement	0.55	0.461–0.647	Moderate

ICC, Intraclass correlation coefficient; CI, confidence interval.

**Table 4 diagnostics-13-01105-t004:** Number of lesions showing specific morphological descriptors on the UF DCE-MRI and conventional DCE-MRI evaluations.

Morphology		Reader 1		Reader 2	
		UF DCE-MRI	Conventional DCE-MRI	UF DCE-MRI	Conventional DCE-MRI
Shape	Round	15	15	11	11
	Oval	120	119	127	124
	Irregular	56	57	53	56
*p*-value		*p* = 1.0		*p* = 0.97	
Margin	Circumscribed	22	13	12	5
	Irregular	152	154	162	162
	Spiculated	17	24	17	24
*p*-value		*p* = 0.17		*p* = 0.13	
Internal enhancement	Homogeneous	23	2	31	3
	Heterogeneous	88	83	77	81
	Rim enhancement	80	106	83	107
*p*-value		*p* < 0.001		*p* = 0.001	

**Table 5 diagnostics-13-01105-t005:** Differences in the morphology between the UF DCE-MRI and conventional DCE-MRI.

Morphology	UF DCE-MRI → Conventional DCE-MRI	Reader 1	Reader 2
Shape	Oval → **Irregular**	4	2
Margin	Circumscribed → **Irregular**	7	10
	Irregular → Spiculated	7	8
Internal enhancement	Homogeneous → **Heterogeneous**	16	16
	Homogeneous → **Rim enhancement**	9	5
	Heterogeneous → Rim enhancement	15	21

The features that helped to obtain a correct diagnosis are shown in **bold**.

## Data Availability

Not applicable.

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
