# Peer review of "Comparison of Ultrafast Dynamic Contrast-Enhanced (DCE) MRI with Conventional DCE MRI in the Morphological Assessment of Malignant Breast Lesions"

_diagnostics, 2023, doi:10.3390/diagnostics13061105_

Round 1

Reviewer 1 Report

In this study, the authors compared the results of the final phase of ultrafast DCEI with the early phase of conventional DCEI in the evaluation of the morphology of malignant breast mass lesions. This comparison was done retrospectively on MRI data of 187 patients (191 lesions) by two radiologists who analyzed morphological descriptors of breast lesions in blinded review. Agreement in lesion evaluations between the radiologists was also statistically analyzed.

The manuscript is clearly written. Goals of the study are well defined and the results are convincing. I have just a few minor comments that are listed below.

1. How fast is ultrafast? Please specify at which conditions one can consider DCEI being ultrafast.

2. In ultrafast DCEI, is the speed advantage gained by reducing spatial resolution or another important image quality parameter?

3. Looking to Table 1 parameters of conventional DCEI and ultrafast DCEI look almost identical. So, it is not clear what makes the difference between these two sequences, i.e., what is the origin for the significant difference in scan time for a phase. Explain also what is the meaning of the phase; does a new image correspond to each phase step?

4. The authors could comment how inclusion of an additional radiologist in evaluation of morphological descriptors of breast lesions between ultrafast and conventional DCEI would influence the results.

5. Could a different contrast agent than gadobutrol be used in the study? If so, what difference would it make? Perhaps with another contrast agent dynamic of accumulation would be different (slower) so that the role of ultrafast DCEI would be diminished.

6. The study was done with two different 3T Siemens scanners. Do other manufacturers (e.g. Philips, GE) use the same type of ultrafast DCEI sequences. How universal is this sequence name? Is it used just for Siemens MRI scanners? If not, then it would be convenient to mention also other commercial names used by other manufacturers.

7. Line 133, The first use of abbreviation ICCs; explain its meaning.

Author Response

Thank you for the thoughtful comments. Per each comment, corresponding text changes are indicated by track changes, comments (R2-1, etc.), and by showing pages and lines.

Point 1: How fast is ultrafast? Please specify at which conditions one can consider DCEI being ultrafast.

Response 1 (R1-1): I specified it in the “Introduction” section. P1, line 43.

“The usual temporal resolution of UF DCE-MRI is less than 10 seconds per frame, which is faster than the 60 seconds per frame of conventional DCE MRI [1].

Point 2: In ultrafast DCEI, is the speed advantage gained by reducing spatial resolution or another important image quality parameter?

Response 2 (R1-2): To continue the previous response sentence, I have explained the UF imaging method from P1, line 45.

“Several accelerated imaging acquisition techniques are available for obtaining high spatial and temporal resolution, including parallel imaging [2, 3], view sharing [4, 5], and compressed sensing (CS) [6, 7]. The CS technique allows high spatial resolution by reconstructing images from randomly and highly under-sampled k-space data acquired with a high temporal resolution [8].

Point 3: Looking to Table 1 parameters of conventional DCEI and ultrafast DCEI look almost identical. So, it is not clear what makes the difference between these two sequences, i.e., what is the origin for the significant difference in scan time for a phase. Explain also what is the meaning of the phase; does a new image correspond to each phase step?

Response (R1-3): Thank you for your suggestion, I distinguished between the time frame and acquisition phase. Please see the Material and methods, 2.2 MR protocols (page 3).

Point 4: The authors could comment how inclusion of an additional radiologist in evaluation of morphological descriptors of breast lesions between ultrafast and conventional DCEI would influence the results.

Response 4 (R1-4): I add the sentence in the “Material and methods,” page 5, “2.3 Image analysis” section, line 170.

“To evaluate the consistency of the visual assessment in terms of morphology,”

Point 5: Could a different contrast agent than gadobutrol be used in the study? If so, what difference would it make? Perhaps with another contrast agent dynamic of accumulation would be different (slower) so that the role of ultrafast DCEI would be diminished.

Point 6: The study was done with two different 3T Siemens scanners. Do other manufacturers (e.g. Philips, GE) use the same type of ultrafast DCEI sequences. How universal is this sequence name? Is it used just for Siemens MRI scanners? If not, then it would be convenient to mention also other commercial names used by other manufacturers.

Response 5, 6 (R1-5, 6): Thank you for the suggestion. I add the sentence in the Discussion, limitation, on page 11, line 410.

“Fourth, the kinetic behavior of gadobutrol may be slightly different from that of other contrast agents [43, 44]. However, we have not been able to examine differences in UF DCE-MRI according to different contrast media or devices. The universal use of this acquisition is a future consideration.”

Point 7: Line 133, The first use of abbreviation ICCs; explain its meaning.

Response (R1-7): Thank you for the suggestion. I decided not to use ICC as an abbreviation.

Reviewer 2 Report

Congratulations on a soundly performed research and your well-written clearly organized and presented and clinically relevant article. It would be interesting if benign lesions were also added in the series. Maybe you can do that in your next work.

1. The authors aim at comparing not only the kinetics, but the shape and characteristics of breast masses as  assessed by ultrafast MRI (UF MRI) compared to dynamic contrast-enhanced MRI and correctly state that no other study has done this in a detailed manner. 

2. Although Zelst et al assessed performance of BI-RADS category between the two modalities in a multireader study, they did not specifically assess accuracy and differences in specific morphological characteristics.  

3. This article includes only mass lesions. The authors don't mention any non-mass like enhancements (NME) and whether/why they excluded NME from the study. If there is a difference in detection/characterization on NMEs, this would seriously hinder clinical utility of UF MRI. Bilateral lesions have been mentioned but none of the patients has been reported to have multifocal/multicentric lesions. This is highly unlikely. Adding data about IHC subtypes of the masses would be helpful. Esp. regarding the discordant masses. Adding benign lesion would have helped improve the impact and quality of the article. But that is only optional. Maybe they can do it in their next study.  

4. Conclusions are relevant and accurate for mass lesions.

5. Tables and figures are clear and representative.

Author Response

Thank you for the thoughtful comments. Per each comment, corresponding text changes are indicated by track changes, comments (R2-1, etc.), and by showing pages and lines.

Point 1: Although Zelst et al assessed performance of BI-RADS category between the two modalities in a multireader study, they did not specifically assess accuracy and differences in specific morphological characteristics.  

Response 2 (R2-1): Thank you for the suggestion. The text has been changed in reference to this point, on page 9, Discussion section, line 337.

“Zelst et al. evaluated the diagnostic performance of UF DCE-MRI breast screening using BI-RADS categories in a multileader study [33], and Dalmis et al. evaluated the diagnostic performance of a multiparametric UF DCE-MRI protocol using artificial intelligence techniques [34]. Although they found that the classification of benign and malignant lesions using UF DCE-MRI was non-inferior to conventional MRI, they did not specifically evaluate differences in accuracy or specific morphologic features. A morphological evaluation is needed when UF DCE-MRI is used for further classification (e.g., between malignant diseases).”

Point 2: This article includes only mass lesions. The authors don't mention any non-mass like enhancements (NME) and whether/why they excluded NME from the study. If there is a difference in detection/characterization on NMEs, this would seriously hinder clinical utility of UF MRI. Adding benign lesion would have helped improve the impact and quality of the article. But that is only optional. Maybe they can do it in their next study.

Response 2 (R2-2): Thank you for the suggestion. I described this on page 8, Discussion, “limitation” section about NME and benign lesions. On page 11, Discussion section, line 398.

“Second, the analysis is inadequate for a clinical situation because we focused only on mass lesions, and non-mass enhancement was excluded, as mentioned in the introduction. A recent study evaluated the size and morphology of ductal carcinoma in situ (DCIS) using UF DCE-MRI with the CS technique. DCIS mostly takes the form of a non-mass lesion, and the study found that the lesion size tended to be smaller and that a clustered ring was not frequently observed on UF DCE-MRI compared to conventional DCE MRI [42]. This result implies that caution is needed when evaluating DCIS on UF DCE-MRI, and that the evidence from the morphological evaluation of mass and non-mass lesions using UF DCE-MRI is insufficient, and further evaluations are needed. ”

Point 3: Bilateral lesions have been mentioned but none of the patients has been reported to have multifocal/multicentric lesions. This is highly unlikely.

Response 3 (R2-3): Thank you for your suggestion. I added the explanation of multiple lesions on page 3, Material and methods section, 2.1 Study population, line 98.

” The study was performed on a per-breast basis. If a single patient had multiple lesions on one side of the breast, the lesion that appeared most malignant or the largest lesion was selected as the index lesion for that breast.”

Point 4: Adding data about IHC subtypes of the masses would be helpful. Esp. regarding the discordant masses.

Response 4 (R2-4): Thank you for your suggestion. I added detail of subtypes in 2.4 histopathological evaluation, and new lines of subtypes in table 2.
